# Enzalutamide Prior to Radium-223 Is Associated with Better Overall Survival in Metastatic Castration-Resistant Prostate Cancer Patients Compared to Abiraterone—A Retrospective Study

**DOI:** 10.3390/cancers15133516

**Published:** 2023-07-06

**Authors:** Hao Xiang Chen, Li-Hsien Tsai, Chao-Hsiang Chang, Hsi-Chin Wu, Ching-Chan Lin, Che-Hung Lin, Chin-Chung Yeh, Chi-Rei Yang, Chi-Shun Lien, Yi-Huei Chang, Ji-An Liang, Guan-Heng Chen, Po-Jen Hsiao, Po-Fan Hsieh, Chi-Ping Huang

**Affiliations:** 1Department of Urology, China Medical University Hospital, China Medical University, Taichung 40402, Taiwan; lylemushroom@gmail.com (H.X.C.); d25792@mail.cmuh.org.tw (L.-H.T.); d8395@mail.cmuh.org.tw (C.-H.C.); d4746@mail.cmuh.org.tw (H.-C.W.); d1436@mail.cmuh.org.tw (C.-C.Y.); d8657@mail.cmuh.org.tw (C.-R.Y.); d12506@mail.cmuh.org.tw (C.-S.L.); d21959@mail.cmuh.org.tw (Y.-H.C.); d20932@mail.cmuh.org.tw (P.-J.H.); 2School of Medicine, College of Medicine, China Medical University, Taichung 40402, Taiwan; d4615@mail.cmuh.org.tw; 3Department of Urology, China Medical University Beigang Hospital, Beigang, Yunlin 651012, Taiwan; 4Department of Internal Medicine, Division of Hematology and Oncology, China Medical University Hospital, China Medical University, Taichung 40402, Taiwan; d13256@mail.cmuh.org.tw (C.-C.L.); d18152@mail.cmuh.org.tw (C.-H.L.); 5Division of Hematology and Oncology, Department of Internal Medicine, An Nan Hospital, China Medical University, Tainan 70965, Taiwan; 6Department of Public Health, College of Public Health, China Medical University, Taichung 406333, Taiwan; 7Department of Radiation Oncology, China Medical University Hospital, Taichung 40402, Taiwan; 8Department of Urology, China Medical University Hsinchu Hospital, Hsinchu 30272, Taiwan; d18149@mail.cmuh.org.tw

**Keywords:** metastatic castration resistant prostate cancer, abiraterone, enzalutamide, radium-223, sequence

## Abstract

**Simple Summary:**

Radium-223 is a bone-targeted radiopharmaceutical that has been shown to improve overall survival (OS) and reduce bone pain in patients with metastatic castration-resistant prostate cancer (mCRPC). In this study, we evaluated the efficacy and safety of radium-223 in patients with mCRPC who had received prior treatment with enzalutamide or abiraterone. We found that patients who received enzalutamide prior to radium-223 had a significantly longer median OS (25.1 months vs. 14.8 months) than those who received abiraterone. We also found that patients who received more than five doses of radium-223 had a longer median OS than those who received fewer doses. These findings suggest that the use of enzalutamide prior to radium-223 and the number of doses of radium-223 received may impact OS in patients with mCRPC.

**Abstract:**

Metastatic castration-resistant prostate cancer (mCRPC) is a progressive stage of prostate cancer that often spreads to the bone. Radium-223, a bone-targeting radiopharmaceutical, has been shown to improve the overall survival in mCRPC in patients without visceral metastasis. However, the impact of prior systemic therapy on the treatment outcome of mCRPC patients receiving radium-223 remains unclear. This study aimed to investigate the optimal choice of systemic therapy before radium-223 in mCRPC patients. The study included 41 mCRPC patients who received radium-223 therapy, with 22 receiving prior enzalutamide and 19 receiving prior abiraterone. The results showed that the median overall survival was significantly longer in the enzalutamide group than in the abiraterone group (25.1 months vs. 14.8 months, *p* = 0.049). Moreover, the number of patients requiring blood transfusion was higher in the abiraterone group than in the enzalutamide group (9.1% vs. 26.3%, *p* = 0.16). The study also found that the number of doses of Radium-223 received was significantly associated with overall survival (≥5 vs. <5, HR 0.028, 95%CI 0.003–0.231, *p* = 0.001). Our study provides insights into the optimal treatment choice for mCRPC prior to radium-223, indicating that enzalutamide prior to radium-223 administration may have better outcomes compared to abiraterone in mCRPC patients without visceral metastasis.

## 1. Introduction

Metastatic prostate cancer inevitably develops castration resistance after androgen deprivation therapy. Castration-sensitive prostate cancer usually responds to androgen deprivation therapy for two to three years before developing castration resistance [1]. Metastatic castration-resistant prostate cancer (mCRPC) has a high mortality rate. Novel hormonal agents (NHA), such as abiraterone and enzalutamide, have greatly prolonged the survival of mCRPC patients [2,3,4].

Patients with mCRPC suffer from symptomatic bone pain, which greatly decreases their quality of life [5]. In addition, spinal cord compression caused by compression fracture secondary to spine metastasis greatly impairs the quality of life of mCRPC patients. Bone health is a concern in this group of patients. Radium-223 is a targeted alpha emitter that binds to bone metastasis locations and effectively reduces symptomatic bone pain. The landmark ALSYMPCA trial showed that radium-223 not only improves bone pain, but it also provides survival benefits to heavily treated mCRPC patients. In addition, it significantly reduces skeletal-related events in mCRPC patients [6]. Studies have shown that Asian populations tend to have a higher proportion of high-volume bone metastasis, which is associated with a prognosis similar to that of visceral metastasis [7].

Currently, radium-223 is indicated for mCRPC patients with symptomatic bone metastasis without visceral metastasis. Radium-223 has been covered by Taiwan’s National Health Insurance since March 2019. In Taiwan, the prescription of radium-223 is strictly limited to symptomatic mCRPC patients who have bone pain with bone metastasis, involving more than two sites and no evidence of visceral metastasis. As a result, there are no restrictions on the number of prior therapy lines that must be exhausted before radium-223 can be prescribed. For symptomatic metastatic castration-resistant prostate cancer (mCRPC) patients, the administration of radium-223 is typically based on the physician’s discretion. The majority of these patients underwent treatment with taxane or NHA, or both, prior to receiving Radium-223.

At present, there are no adequate data regarding the impact of treatment prior to radium-223 on a patient’s survival. The optimal choice of systemic treatment prior to radium-223 is unknown. Currently, the optimal sequencing of mCRPC is only presented in terms of first-line, second-line, and third-line settings. The optimal agents to use before radium-223 are unknown, as there are different NHAs available in the market [8]. The goal of this study is to retrospectively investigate the impact of different NHAs on mCRPC patients receiving radium-223. Our study may provide insight into the optimal choice of NHA in mCRPC patients prior to Radium-223.

## 2. Methods

We conducted a retrospective study on all mCRPC patients who received radium-223 treatment at a tertiary referral center from 1 October 2019 to 22 September 2022. Patients with incomplete information on their previous prostate cancer treatment, those who were enrolled in other trials, those who had been exposed to more than one NHA, or those who had double cancer were excluded from the study. The Institutional Review Board of China Medical University Hospital, Taichung, Taiwan, approved the study (Protocol Number: CMUH109-REC1-082).

Patients were separated into two groups based on the NHA they received prior to radium-223: enzalutamide or abiraterone. Prior docetaxel treatment was classified as ‘early’ if it was administered during the castration-sensitive phase of prostate cancer, whereas it was classified as ‘late’ if it was given during the metastatic castration-resistant prostate cancer (mCRPC) phase. The primary outcome was the overall survival of patients receiving radium-223, defined as the time from the first radium-223 dose to death. Secondary outcomes included median skeletal-related event (SRE) free survival, median time to increase in total alkaline phosphatase (Alk-p) (defined as two consecutive increase of ≥25% from the baseline in patient with no ALP-p decrease or above nadir in those with ALK-p decrease), median time to increase in prostate-specific antigen (PSA)) (defined as two consecutive increase of ≥25% from the baseline in patient with no PSA decrease or above nadir in those with PSA decrease), number of patients with >30% reduction of Alk-p, number of patients with abnormal Alk-p at baseline resulting in the normalization of Alk-p after receiving radium-223, and hematologic adverse events, such as anemia and thrombocytopenia, as well as the number of patients requiring blood transfusion. SRE was defined as the first use of external-beam radiation therapy to relieve skeletal symptoms, new symptomatic pathologic vertebral or nonvertebral bone fractures, spinal cord compression, or tumor-related orthopedic surgical intervention, based on the ALYMPCA trial. The median time to castration-resistant prostate cancer (CRPC) was defined as the duration from the initial diagnosis of prostate cancer to the onset of castration resistance. CRPC refers to prostate cancer that progresses clinically, radiographically, or biochemically, despite the presence of castrate levels of serum testosterone (<50 ng/dL). The median time from CRPC to Radium-223 was defined as the interval between the initiation of radium-223 treatment and the diagnosis of CRPC.

We collected patient data from electronic medical records and rounded percentage values to two decimal points. Overall survival analysis was conducted using the log-rank test, while the Student T-test and the Chi-square test were used for continuous and categorical data analysis, respectively. We used a multivariate Cox proportional hazard model to evaluate prognostic factors, and statistical analysis was performed using IBM SPSS Statistics 25. A *p*-value of less than 0.05 was considered statistically significant.

## 3. Results

Between 1 October 2019 to 22 September 2022, a total of 60 patients with mCRPC were treated with radium-223. We excluded nineteen patients for the following reasons: seven patients had no prior NHA use, six patients had received both abiraterone and enzalutamide, four patients were enrolled in other clinical trials, one patient had incomplete data, and one patient was diagnosed with concurrent hepatocellular carcinoma. After exclusions, 41 patients were included for analysis. Of these, 22 patients had received enzalutamide, and 19 patients had received abiraterone prior to treatment with radium-223. The median age of patients in the enzalutamide group was 75.5 years (range 64–89) and 76 years (range 53–94) in the abiraterone group. The median follow-up time was 28.2 months (range 1.5–39.9). More patients in the enzalutamide group were able to receive more than five doses of radium-223 compared to the other group. Specifically, 77.3% of patients in the enzalutamide group received more than five doses, while only 57.9% in the abiraterone group received the same. The proportion of patients with docetaxel exposure and indications of docetaxel prior to Radium-223 were similar between the two groups (Table 1). Docetaxel was given after mCRPC diagnosis, rather than being used upfront in most patients (76.9% in the enzalutamide group and 92.3% in the abiraterone group). The initial biochemistry values at the start of radium-223 therapy are summarized in Table 1. Baseline hemoglobin, absolute neutrophil count, ALK-P, LDH, and PSA were similar between the two groups, except for the mean platelet count, which was higher in the enzalutamide group compared to the abiraterone group.

At the time of data collection, 22 patients had died, with 10 out of 22 (45.5%) being in the enzalutamide group and 12 out of 19 (63.2%) being in the abiraterone group. The median survival time for the enzalutamide group was 25.1 months (95% CI 19.1–31.1), and for the abiraterone group, it was 14.8 months (95% CI 9.5–20.0), as shown in Table 2. A Kaplan-Meier curve showed that the median overall survival of patients in the enzalutamide group was significantly longer than that of the other group (*p* = 0.049) (Figure 1A). The mean duration of exposure to radium-223 was also significantly longer in the enzalutamide group (5.0 vs. 3.9 months, *p* = 0.02).

In each group, there were 3/22 (13.6%) SRE in the enzalutamide group and 2/19 (10.5%) SRE in the abiraterone group. The SRE experienced by these patients were new bone fractures that required surgical intervention and external beam radiation therapy for the relief of skeletal symptoms (Appendix A). The median SRE-free survival for the enzalutamide and abiraterone groups was 8.5 months and 6.6 months, respectively, after the initiation of the first dose of radium-223. Bone pain improvement was observed in both groups, and the median dose for reducing opioid use was the second dose in both groups. The number of SRE, median SRE-free survival, and median dose to de-escalation of analgesic were similar between the two groups (Table 2).

The biochemical outcomes are summarized in Table 3 and Table 4. Overall, hemoglobin, platelet, and alkaline phosphatase decrease after radium-223 administration, while PSA levels continue to rise in both groups. The changes in these biochemical values are similar between the two groups. The mean change in hemoglobin after the last cycle of radium-223 is −1.38 mg/dL and −1.0 mg/dL in the enzalutamide and abiraterone groups, respectively. For Alk-p, the mean changes are −11.36 U/L and −55 U/L, respectively. Most of the patients in this study had some degree of anemia. In the enzalutamide group, 21 patients (95.4%) had some degree of anemia during radium-223 treatment, and most of the anemia was self-limited, asymptomatic, and did not require intervention. However, 2/22 (9.1%) of the patients with anemia in the enzalutamide group required blood transfusion during the radium-223 treatment course. For the abiraterone group, the number of patients requiring blood transfusion was 5/19 (26.3%).

The median time to CRPC for all patients was 24.8 months (range 4.5–164.9), while the median time from CRPC to Radium-223 was 24.9 months (range 1.5–85.7). A Kaplan-Meier curve showed that the number of doses greater than five has significantly longer median overall survival, 25.5 months (95% CI 23.4–33.7) vs. 5.8 months (95% CI 3.8–7.9), *p* < 0.001 (Figure 1B). The mOS of the group with time to CRPC lower than median vs. greater than the median was 18.9 months (95% CI 12.9–24.9) vs. 23.1 months (95% CI 16.4–29.7), *p* = 0.69 (Figure 1C). The mOS for the group of the time from CRPC to Rad-223 lower than median vs. greater than median was 17.8 months (95% CI 11.9–23.6) vs. 25.0 months (95% CI 18.1–31.9), *p* = 0.298 (Figure 1D). Figure 1E showed that the mOS for prior docetaxel vs. docetaxel-naïve was 21.7 months (95% CI 14.7–28.6) vs. 19.6 months (95% CI 13.7–25.6), *p* = 0.82). Further analysis revealed a significantly longer median overall survival in the enzalutamide-complete group compared to the enzalutamide-incomplete group (29.5 months vs. 5.3 months, *p* < 0.0001), as well as a significantly longer overall survival in the abiraterone-complete group compared to the abiraterone-incomplete group (22.8 months vs. 4.0 months, *p* = 0.001) (Appendix A).

Figure 2 presents the findings of the Cox proportional hazard model analysis, which aimed to investigate the impact of various risk factors on the median overall survival of mCRPC patients who received NHA and radium-223 treatment. The analysis revealed that patients who were administered enzalutamide (HR 0.223, 95% CI: 0.062–0.810, *p* = 0.023) and those who received doses of five or more (HR 0.028, 95% CI: 0.003–0.231, *p* = 0.001) had significantly better median overall survival rates. On the other hand, the study did not find any significant association between prior docetaxel exposure, time from CRPC to Radium-223, time to CRPC, the severity of bone metastasis, and the median overall survival of mCRPC patients receiving NHA and radium-223 treatment.

## 4. Discussions

In this retrospective study, we found that administering enzalutamide prior to radium-223 was significantly associated with longer mOS compared to abiraterone. Multivariate analysis demonstrated that the use of enzalutamide and the number of doses of radium-223 were also significantly associated with better mOS. The study population consisted of mCRPC patients who had received one NHA prior to radium-223, with over half of the included patients having prior docetaxel exposure. The proportion of patients with prior chemotherapy exposure was higher in our study than in other studies. In a study by P. Jarvis et al., the median overall survival for mCRPC patients who had received one prior systemic therapy was 665 days [9]. In the study by M. Smith et al., discussing the combination of abiraterone and radium-223, the mOS was 30.7 months in the combination groups and 33.3 months in the abiraterone groups [10]. However, only 2% of the patients in that study had received prior docetaxel, which is much lower than the percentage of patients with prior docetaxel exposure in our study. The mOS of the abiraterone group in our study was only 14.8 months, which was much shorter than in other studies.

The timing of radium-223 administration may influence its overall survival benefit. To determine whether radium-223 was given early or late, we used the time from CRPC to radium-223 as a surrogate marker. The time from CRPC to radium-223 in our study was much longer than in other studies. In our study, the median time from CRPC to the first dose of radium-223 in the enzalutamide group was 25.3 months, and it was 24.9 months in the abiraterone group. In the study by DJ George et al., the median time from mCRPC to radium-223 was around 10 months [11]. In D.J. George et al.’s study, the median time from mCRPC to radium-223 for patients with survival times greater than two years was only 7.1 months. This suggests that administering radium-223 earlier is correlated with longer survival times for patients. In Jiang et al.’s study, more than 70% of the patients received radium-223 in first- or second-line settings. In our study, most of the patients were administered radium-223 in the third-line setting. The time from CRPC to radium-223 was similar between the two groups in our study, and both were longer than in other studies. Therefore, our findings suggest that the patient in our study received radium-223 in a much later stage of their disease. This may explain the difference in our results in comparison to the other studies.

Multivariate analysis showed that only enzalutamide and greater than five doses of radium-223 were associated with significantly lower mortality. An amount of 77.3% of the patients who had prior enzalutamide were able to complete five or six doses of radium-223, while only 57.9% of the patients in the abiraterone group were able to do so. The number of doses of radium-223 received was found to be correlated with better overall survival in previous studies [11,12,13]. Jiang XY et al. showed that the number of radium-223 doses was correlated with better survival outcomes [12]. In that study, the mOS for patients who received five to six doses of radium-223 was 15.8 months, compared to 4.7 months (*p* = <0.001) for those who received only one to four doses. A meta-analysis also found that enzalutamide is associated with longer radiological progression-free survival compared to abiraterone [14].

It is postulated that the effect of these NHAs on the bone microenvironment is related to the treatment effect of radium-223. Radium-223 works by emitting alpha-particles after incorporating into the bone matrix, and the bone microenvironment may have played a role in its efficacy [15]. Abiraterone is always administered with prednisolone concomitantly in our patients. Glucocorticoids are known to increase bone resorption by modulating the RANK/RANKL/OPG pathway [16,17,18]. In addition, glucocorticoids are known to suppress bone formation by directly inhibiting osteoblast proliferation [19]. The effects of enzalutamide on the bone microenvironment are not yet available in the literature, and they are worthy of further investigation. Therefore, we hypothesize that the effect of enzalutamide and abiraterone + prednisolone on the bone microenvironment may be related to our finding that the enzalutamide group received a higher number of doses of radium-223, which was correlated with significantly longer overall survival.

Several studies have attempted to identify the impact of treatment sequencing in mCRPC. However, this issue remains controversial, as some studies show positive findings, while others do not. It is intriguing to know the optimal sequence in mCRPC treatment. Currently, there is no recommended treatment sequence for radium-223, and it is unclear whether prior systemic therapy affects treatment outcomes for mCRPC patients receiving radium-223. One network meta-analysis by Junru Chen et al. compared abiraterone, enzalutamide, cabazitaxel, and radium-223 in mCRPC patients who had progressed after docetaxel treatment [20]. This network meta-analysis revealed that following docetaxel failure, enzalutamide demonstrated the most substantial improvements in overall survival and progression-free survival when compared to abiraterone, cabazitaxel, and radium-223. These findings are based on data extracted from clinical trials COU-AA-301, AFFIRM, TROPIC, ALSYMPCA, and a study conducted by Sun et al. [21,22,23,24].

A prospective trial conducted by Jiang X.Y. et al. (N = 296) demonstrated that patients without prior docetaxel treatment experienced superior overall survival compared to those with prior docetaxel treatment (12.3 months vs. 8.1 months, *p* = 0.022) [12]. Additionally, a retrospective study by Chiang et al. reported similarly significant findings. In their study involving 48 patients with a mean follow-up period of 12.4 months, patients without prior docetaxel exhibited a considerably lower all-cause mortality rate compared to those with prior docetaxel treatment (40.0% vs. 78.3%, *p* = 0.02) [25]. More than half of the patients in our study had prior docetaxel, while there was only 26.3% in Jiang X.Y. et al.’s study [12]. Multivariate analysis of our study showed that the hazard ratio of prior docetaxel was 3.46 (95% CI 0.92–13.04), which is in line with the previous studies. Another retrospective study by Oguma et al. of 64 patients yielded results similar to our study, indicating that administering abiraterone (HR 2.530, 95%CI 1.163–5.502, *p* = 0.019) prior to radium-223 was associated with worse overall survival, while enzalutamide (HR 1.425, 95%CI 0.685–2.967, *p* = 0.343) was not associated with worse overall survival [26]. The proportion of patients with prior docetaxel in this study was 50%, which is similar to our study population [26]. These findings support our conclusion that administering abiraterone prior to enzalutamide in the latter treatment course is associated with worse overall survival. It should be noted that the study by Oguma et al. did not classify patients into subgroups based on their prior use of abiraterone or enzalutamide, and it also failed to report critical information, such as the time from CRPC to radium-223 treatment, the sequence of prior therapies, and the extent of the disease. The heterogeneous results of these studies may be due to the significant variation in study populations. Besides, most of these studies are retrospective studies in which the treatment was tailored to each patient clinically. Therefore, these studies are prone to selection bias.

Most previous studies have focused on the effects of combining NHA with radium-223. However, these studies have shown that combination therapy does not result in better overall survival compared to NHA alone [10,27,28,29,30]. The ERA223 trial showed that combining abiraterone with radium-223 not only fails to improve patient survival, but it also increases skeletal events [10]. Therefore, the combination use of abiraterone and radium-223 is not recommended. In a study investigating the combination of radium-223 with abiraterone or enzalutamide, the combination therapy did not improve overall survival [28]. Furthermore, treatment-related adverse events were found to increase in combination therapy. In our study, the incidence of skeletal-related events (SREs) was comparable between both groups and notably lower compared to findings from other studies. Specifically, the SRE rate in the enzalutamide group was 18.7%, while, in the abiraterone group, it was 13.3%. In the ALSYMPCA trial, the SRE rates were 33% in the Radium-223 group and 38% in the placebo group [6]. Similarly, in the ERA 223 trial, the SRE rates were 49% in the radium-223 group and 47% in the placebo group [10]. It is possible that SREs are only increased with combination use but not sequential use. Another study conducted by Neal Shore et al. reported comparable outcomes between concurrent treatment of NHA and radium-223 and a layered treatment approach [31]. In their study, concurrent treatment was defined as the administration of both NHA and radium-223 within a 30-day period, while the layered approach involved administering one drug more than 30 days after the first. In contrast, our study focuses on a different treatment approach, where radium-223 was administered after the discontinuation of NHA. This means that none of the patients in our study received concurrent treatment of radium-223 and NHA. Instead, all patients in our study received ADT + radium-223 after NHA treatment failure. As previous studies have focused on the outcomes of combining NHA with radium-223, rather than the treatment sequence, our study provides insight into the optimal treatment choice in mCRPC.

Most of the patients in this study experienced some degree of anemia during their course of radium-223 treatment. The mean change in hemoglobin levels in our study was −1.38 g/dL in the enzalutamide group and −1.01 g/dL in the abiraterone group, which is comparable to the results of studies involving radium-223 alone or in combination therapy [32,33]. However, the rate of anemia was higher in our study compared to other studies [34,35,36]. This suggested that the patients in our study had more extensive bone marrow involvement. The number of patients requiring blood transfusions in the abiraterone group was higher than in the enzalutamide group, although this difference was not statistically significant. Our study results suggest that enzalutamide given prior to radium-223 is associated with a longer median overall survival and possibly less severe hematologic adverse events. As radium-223 is indicated in mCRPC patients without visceral metastasis, enzalutamide may be a better choice for mCRPC patients with symptomatic bone metastasis without visceral metastasis, as it may result in longer overall survival with lower hematologic toxicity.

The main limitation of our study is its retrospective nature, which means that some data related to treatment, such as minor treatment-related adverse events and quality of life, were not fully available. Furthermore, the number of enrolled patients in this study is relatively small. Our study could not investigate the impact of darolutamide and apalutamide prior to radium-223, as they were not covered by national health insurance in Taiwan at the time of radium-223 initiation.

However, our study is the first to suggest that the use of NHA prior to radium-223 may affect the survival of mCRPC patients. Most clinical trials have focused on investigating the effect of combining radium-223 with NHA, while our study provides insight into the optimal choice of systemic therapy before radium-223 in mCRPC patients. Therefore, a prospective study with a larger population is needed to confirm the findings of our study.

## 5. Conclusions

In conclusion, our study suggests that enzalutamide may be a better choice than abiraterone prior to radium-223 administration in mCRPC patients, as the overall survival is significantly longer in the enzalutamide group.

## Figures and Tables

**Figure 1 cancers-15-03516-f001:**
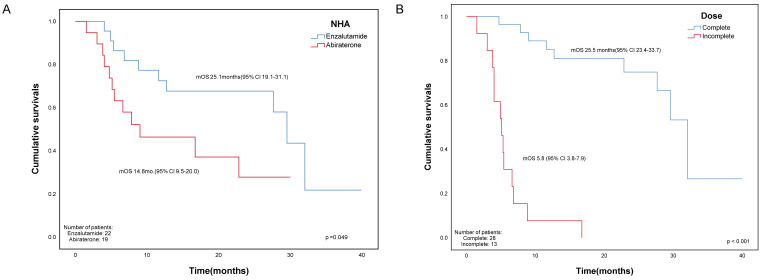
The Kaplan-Meier curve for different subgroups. (**A**) The mOS of enzalutamide versus abiraterone showed a significantly longer mOS in the enzalutamide group. (**B**) Complete (≥5 doses) versus incomplete (<5 doses) showed significantly longer mOS in the complete group. (**C**) Time to CRPC, greater than the median time versus lesser than the median time. The median time to CRPC was 24.8 months (range 4.5–164.9) (**D**) The time from CRPC to Radium-223, greater than median time versus lesser than median time, and the median time from CRPC to Radium-223 was 24.9 months (range 1.5–85.7). (**E**) Docetaxel status comparing prior docetaxel for mCRPC versus those without. Abbreviations: NHA: novel-hormone agent, mOS: median overall survival, CRPC: castration-resistant prostate cancer, rad-223: radium-223, CI: confidence interval.

**Figure 2 cancers-15-03516-f002:**
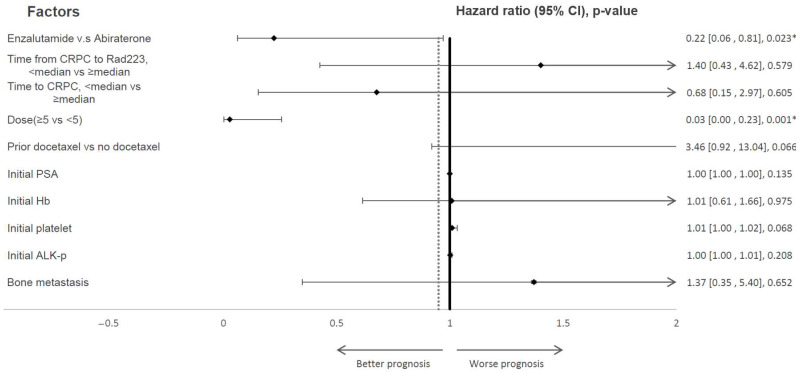
The forest plot illustrates the multivariate analysis for predicting the overall survival in patients with mCRPC who received radium-223 treatment. The use of enzalutamide prior to radium-223 and the administration of more than five doses of radium-223 are significantly associated with improved overall survival in mCRPC patients. Abbreviations: CRPC: castration-resistant prostate cancer; NHA: novel hormonal agent; ALK-P: alkaline phosphatase; PSA: prostate-specific antigen, Hb: hemoglobin. * ≤ 0.05.

**Table 1 cancers-15-03516-t001:** Baseline characteristics.

Parameters	Enzalutamide	Abiraterone	*p*-Value
Number of patients	22	19	
Age			
Median(range)-year	75.5(64–89)	76(53–94)	0.44
>75 year (%)	11(50)	11(57.9)	0.61
Prior docetaxel (%)	13(59.1%)	13(68.4)	0.55
Early	3(23.1)	1(7.7)	
Late	10(76.9)	12(92.3)	
Dose of Radium-223 (%)			
≥5	17(77.3)	11(57.9)	0.24
3–4	4(18.2)	4(21.1)	
≤2	1(4.5)	4(21.1)	
Prior denosumab (%)	13(59.1)	13(68.4)	
Median time to CRPC, mo. (range)	24.6(7–164.9)	26.9(4.5–78.4)	0.56
Median time from CRPC to Radium-223, mo. (range)	25.3(1.5–75)	24.9(2.9–85.7)	0.88
Median Duration of NHA	13.1(5.0–61.2)	11.0(0.2–50.2)	0.78
Total ALK-p			
<220	21(95.5)	16(84.2)	0.23
≥220	1(4.5)	3(15.8)	
Extent of disease			0.855
<6 metastases	5(22.7)	3(15.8)	
6–20 metastases	9(40.9)	7(36.8)	
>20 metastases	8(36.4)	7(36.8)	
Superscan *	0(0)	2(10.5)	
Use of analgesic	18(81.8)	16(84.2)	0.84
Use of opioid	8(36.4)	10(52.6)	0.29
Baseline median Biochemical value (range)			
Hb, g/dL	12.4(8.2–14.6)	11.6(7.8–12.9)	0.12
Platelet, 10^3^/µL	212(120–403)	155.0(72.0–228.0)	0.04
ANC, 10^3^/µL	4035(1053–10896)	4316(1938–4954)	0.25
Albumin, g/dL	4(3.7–4.3)	3.6(2.3–3.8)	0.15
Total ALK-P, U/L	89(47–717)	165(61–1109)	0.32
LDH, U/L	171(111–372)	205(127–346)	0.54
PSA, ng/mL	76.1(7.08–2550.6)	387.0(4.5–4557.6)	0.19

* Superscan: the bone scan showing the intense and diffuse uptake of the skeletal tracer, Abbreviations: NHA, novel-hormonal agents; CRPC, castration-resistant prostate cancer; LDH, lactate dehydrogenase.

**Table 2 cancers-15-03516-t002:** The primary and secondary endpoints.

Endpoints	Enzalutamide	Abiraterone	*p*-Value
Primary			
mOS from the first dose of Rad223, mo.	25.1	14.8	0.049 *
Secondary			
mean Duration of exposure, mo.	5.0	3.9	0.02 *
Alk-p-decrease	10(45.4)	8(42.1)	0.83
>30%	8(80)	8(100)	0.48
Normalization (%)	5(50)	7(87.5)	0.93
Median time to Alk-p increase, mo.	3.1	2.6	0.92
PSA decrease	3	2	0.76
PSA increase	14	11	0.71
Median time to PSA increase	44	22	0.83
SRE (%)	3(18.7)	2(13.3)	0.76
Median SRE-free survival	8.5	6.6	0.96
De-escalation of analgesic	10(55.5)	7(43.8)	0.49
Median dose to de-escalation	2	2	
Anemia	21(95.4)	19(100)	1.0
Blood transfusion	2(9.1)	5(26.3)	0.16

PSA increase defined as time to PSA increase from the date of the first Radium223. Abbreviations: mOS: median overall survival, SRE: skeletal-related events. * ≤0.05.

**Table 3 cancers-15-03516-t003:** The mean hematologic and biochemistry data for cycle 1 and the last cycle.

Values, Mean ± SD	Enzalutamide		Abiraterone	
	Cycle 1	Last Cycle	Cycle 1	Last Cycle
Hemoglobin, g/dL	12.1 ± 1.8	10.8 ± 1.7	11.4 ± 1.2	10.4 ± 1.8
Platelet, 10^3^/µL	233.1 ± 71.8	202.0 ± 73.0	180.2 ± 62.5	154.9 ± 102.1
ALK-P, U/L	133.9 ± 149.9	117.0 ± 99.1	207.4 ± 255.9	128.2 ± 205.6
PSA, ng/mL	253.8 ± 574.6	903.0 ± 1519.4	672.6 ± 1196.3	1764.7 ± 3717.5

Abbreviations: ALK-P: alkaline phosphatase; PSA: prostate-specific antigen.

**Table 4 cancers-15-03516-t004:** Mean change of the laboratory data.

Values, Mean ± SD	Enzalutamide	Abiraterone
Hemoglobin, g/dL	−1.4 ± 1.04	−1.0 ± 1.7
Platelet, 10^3^/µL	−31.1 ± 67.1	−25.2 ± 89.6
ALK-P, U/L	−11.4 ± 96.7	−55.0 ± 83.3
PSA, ng/mL	654.4 ± 1041.7	1126.8 ± 2746.8

Abbreviations: ALK-P: alkaline phosphatase; PSA: prostate-specific antigen.

## Data Availability

The data presented in this study are available on request from the corresponding author.

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
