# Peer review of "Enzalutamide Prior to Radium-223 Is Associated with Better Overall Survival in Metastatic Castration-Resistant Prostate Cancer Patients Compared to Abiraterone—A Retrospective Study"

_cancers, 2023, doi:10.3390/cancers15133516_

Round 1

Reviewer 1 Report

The authors conducted a careful review of clinical data from 60 patients and reached an encouraging conclusion that the administration of Radium-223 following Enzalutamide treatment was both safe and beneficial for mCRPC patients. However, there are some aspects where the data and figures could be further improved and additional information could be provided:

It would be valuable to include related data on skeletal-related events (SRE) in mCRPC patients, such as the median symptomatic skeletal event-free survival and the number of patients experiencing at least one symptomatic skeletal event or death. The details of these events, including external beam radiotherapy, symptomatic pathological bone fractures, spinal cord compression, and death, could be included.

The Phase III ERA223 trial concluded that concurrent treatment with abiraterone acetate plus prednisone and radium-223 did not improve symptomatic skeletal event-free survival in patients. It would be helpful to discuss the observations from this study and compare them to the findings of the current research. Specifically, it would be informative to understand the extent to which radium-223 improved skeletal event-free survival in the patients studied.

The study by Neal Shore et al. (Prostate Cancer and Prostatic Diseases, 2020) concluded that there was no difference between concurrent and layered treatment. It would be important to address this finding and reconcile it with the conclusions of the current study. Exploring potential reasons for the differences in results and discussing the implications would enhance the overall significance.

The multivariate analysis in Table 5 suggests that prior docetaxel treatment may be beneficial to patients (p = 0.066), while the p-value in Table 1 is reported as 0.55. It is essential to address this disparity and provide an explanation for the inconsistent findings. Furthermore, discussing the potential impact of a larger cohort study when comparing patients who received prior docetaxel versus those who did not would add depth to the discussion.

The description of the treatment sequence from line 243 to line 267 could be improved. For example, for the study by Junru Chen et al., the treatment sequence was not clearly mentioned, but it was restated that Enzalutamide was superior to other treatments. Providing a more detailed and organized account of the treatment sequence, particularly in comparison to other studies, would enhance clarity and reader understanding.

Minor points for improvement:

The alignment of parameters in each table could be adjusted for better readability. For instance, the "Early" and "Late" under "Prior docetaxel" should be properly indented.

It would be beneficial to clearly indicate the primary endpoint and secondary endpoints in Table 2 to provide a concise overview of the study's objectives.

The meaning of "Extent of disease" in Table 1 should be noted below the table or explained in the main text to ensure clarity for readers.

The classification of hormonal therapy as systemic therapy should be clarified. While Taxane drugs are typically considered systemic therapy, the categorization of Enzalutamide and Abiraterone as hormonal therapy may require further explanation or justification.

For patients who received prior Enzalutamide or Abiraterone, it would be helpful to provide information on the time interval between Radium-223 administration and the previous treatments. Additionally, providing a rationale for the administration of Radium-223 and presenting it in a schematic plot format would enhance readability and highlight the novelty of this study.

Converting the data from Table 5 into a hazard ratio plot would provide a visual representation of the results and aid in understanding the impact of different variables on the study outcomes.

Author Response

Please note that all the number of lines mentioned in this response to the reviewer correspond to the lines in tracking mode with simple markup.

Reviewer 2 Report

1. please clarify how many lines of therapy are necessary before Radium-223 is allowed to be prescribed in the mCRPC setting in Taiwan, the EMA has restricted access to Radium-223 to patients who have failed at least 2 lines of therapy including chemo or are not fit for chemo.

2. how many patients had chemotherapy before Radium-223 and what had been the treatment sequencing for eligible patients in the mHSPC setting, did any patients received chemo in that setting?

3. did all patients have solely bone metastases?

4. where all patients under bone protective agents and if not why?

5. how do you account for the difference in outcome between the two studied treatment options on the basis of biology?

6. what was the rationale for choosing Radium-223 over other options in the mCRPC setting? physician's choice, local guidelines, other?

the quality of English is good pending some minor grammatical and syntaxis errors.

Author Response

(The authors gave the same response as above.)

Reviewer 3 Report

Summary:

Androgen deprivation therapy (ADT) is the first line treatment for metastatic castration-sensitive prostate cancer (mCSPC). However, metastatic castration-resistant prostate cancer (mCRPC) inevitably relapses in most of the patients. The development of novel hormonal agents (NHA), such as enzalutamide and abiraterone, has greatly prolonged the overall survival of patients with mCRPC, but the life quality of the patients dramatically decreased due to the symptomatic pain resulted from visceral metastasis or/and bone metastasis. Radium-223 is a targeted alpha emitter that effectively reduces bone pain for mCRPC patients with bone metastasis but without visceral metastasis. It is generally administered after the patient has been treated with taxane or NHA, or both. However, the optimal agents used before radium-223 remain unknown as there are various options of NHA. To investigate the optimal choice of systemic therapy before radium-223 for mCRPC patients, Haoxiang Chen, et al retrospectively studied 41 mCRPC patients who received radium-223 therapy, with 22 of them following enzalutamide treatment and 19 of them following abiraterone treatment. The results showed that either the use of enzalutamide prior to radium-223 therapy or 5 doses or more of radium-223 was significantly associated with higher overall survival, compared to the use of abiraterone or lower than 5 doses of radium-223. This manuscript is well structured with sufficient background in the introduction, adequately described methods, and appropriate research design. However, I have several comments on the result presentation and conclusions.

Major points:

(1) The authors plotted the Kaplan-Meier curve for Enzalutamide vs Abiraterone (Figure 1A) regardless of the number of radium-223 dose and for Complete dose vs Incomplete dose (Figure 1B) regardless of the use of NHA prior to radium-223 therapy. Since only these two parameters showed impact to the overall survival of the mCRPC patients, I suggest that the authors further specify the systemic therapy before radium-223 to compare the overall survival of the mCRPC patients in following subgroups: Enzalutamide+Incomplete dose vs Enzalutamide+Complete dose vs Abiraterone+Incomplete dose vs Abiraterone+Complete dose.

(2) The authors plotted the Kaplan-Meier curve for Less than median time to CRPC vs Longer than median time to CRPC (Figure 1C) and Less than median time from CRPC to Rad-233 vs Longer than median time from CRPC to Rad-233 (Figure 1D) regardless of the use of NHA and the number of radium-223 dose. The authors should show the median time of the time to CRPC and of the time from CRPC to Rad-233 among all patients in the table, and show the number of patients in the subgroups of Figure 1C and Figure 1D in the table.

(3) In the Conclusion (Line 308 to Line 310), the authors claimed that the use of enzalutamide prior to radium-223 therapy is associated with less hematologic toxicity compared to the use of abiraterone, but the data related to hematologic toxicity, such as anemia and blood transfusion, do not show significant difference between enzalutamide-treated group and abiraterone-treated group.

Author Response

(The authors gave the same response as above.)
